# Changes in Cancer Care for Patients Aged 80 and Above: A Cohort Study from Samsung Comprehensive Cancer Center in South Korea

**DOI:** 10.3390/cancers17122017

**Published:** 2025-06-17

**Authors:** Seung Tae Kim, Danbee Kang, Seok Jin Kim, Jun Ho Lee, Hong Kwan Kim, Yong Beom Cho, Yong Han Paik, Seok Won Kim, Byong Chang Jeong, Ho Jun Seol, Man Ki Chung, Kyu Taek Lee, Kihyun Kim, Sung-wook Seo, Jeong-Won Lee, Hee Chul Park, Dong Wook Shin, Juhee Cho, Won Kim, Jeeyun Lee, Woo Yong Lee

**Affiliations:** 1Management & Supportive Office, Seoul 06355, Republic of Korea; seungtae1.kim@samsung.com (S.T.K.); seokjin88.kim@samsung.com (S.J.K.); 2Clinical Epidemiology Center, Seoul 06355, Republic of Korea; dbee.kang@samsung.com; 3Gastric Cancer Center, Seoul 06355, Republic of Korea; junho87.lee@samsung.com; 4Lung and Esophageal Cancer Center, Seoul 06355, Republic of Korea; hkts.kim@samsung.com; 5Colorectal Cancer Center, Seoul 06355, Republic of Korea; yongbeom.cho@samsung.com; 6Liver Cancer Center, Seoul 06355, Republic of Korea; yh.paik@samsung.com; 7Breast Cancer Center, Seoul 06355, Republic of Korea; seokwon1.kim@samsung.com; 8Genitourinary Cancer Center, Seoul 06355, Republic of Korea; bc2.jung@samsung.com; 9Brain Tumor Center, Seoul 06355, Republic of Korea; hojun.seol@samsung.com; 10Head and Neck Cancer Center, Seoul 06355, Republic of Korea; manki.chung@samsung.com; 11Pancreatobiliary Cancer Center, Seoul 06355, Republic of Korea; ktcool.lee@samsung.com; 12Hematologic Malignancy Center, Seoul 06355, Republic of Korea; kihyunk.kim@samsung.com; 13Rare Cancer Center, Seoul 06355, Republic of Korea; sungwook.seo@samsung.com; 14Gynecologic Cancer Center, Seoul 06355, Republic of Korea; garden.lee@samsung.com; 15Proton Therapy Center, Seoul 06355, Republic of Korea; hee.ro.park@samsung.com; 16Supportive Care Center, Seoul 06355, Republic of Korea; dongwook.shin@samsung.com; 17Cancer Education Center, Seoul 06355, Republic of Korea; jh1448.cho@samsung.com; 18CAR T-Cell Therapy Center, Seoul 06355, Republic of Korea; wonseog.kim@samsung.com; 19Precision Cancer Therapeutics Center, Seoul 06355, Republic of Korea; jyun.lee@samsung.com; 20Samsung Comprehensive Cancer Center, Samsung Medical Center, Sungkyunkwan University School of Medicine, Seoul 06355, Republic of Korea

**Keywords:** elderly patients, geriatric oncology, survival, treatment trends, real-world data

## Abstract

With 70% of new cancers expected to be diagnosed in older adults within a decade, cancer care for this population has gained increasing global attention. Findings from this cohort of the SMC Cancer Registry highlight key trends, including a rising number of patients aged ≥80 years and an increasing proportion receiving treatment—particularly after 2020, when more than 60% received therapy. Furthermore, survival benefits associated with treatment were comparable to those observed in younger patients across all cancer types.

## 1. Introduction

With 70% of new cancers expected to be diagnosed in older adults within a decade, cancer care for this population has attracted increasing global attention [1]. Older adults often present with comorbidities, frailty, reduced baseline life expectancy, and potential differences in tumor biology [2]. As a result, conventional cancer treatments pose a risk of overtreatment, exposing older patients to unnecessary toxicity and compromising their quality of life [3].

A significant proportion of older adults are less likely to receive optimal cancer treatments. According to a prior study, only 30% of patients aged 80 years and older received chemotherapy, compared to 65% of those aged 18–59 years [4]. However, recent studies suggest that older adults in good functional health can still derive meaningful survival benefits from active cancer treatment, particularly when treatment decisions are based on frailty rather than chronological age [5]. In fact, recent improvements, in general, health, driven by better nutrition, healthier lifestyles [6], and expanded access to cancer screening, have contributed to earlier detection. This has led to a growing subset of older patients who are able to undergo and benefit from active treatment [7,8]. Additionally, innovations such as immune checkpoint inhibitors and targeted therapies are transforming cancer care, providing lower toxicity and improved tolerability [8,9,10]. This shift has generated greater optimism that older adults can benefit from more active treatments.

Despite these advancements, geriatric oncology remains challenging in real-world settings due to limited evidence. Previous studies have focused on a narrow set of cancers, such as breast, colorectal, and lung cancer, and have lacked stage stratification. Moreover, prior research has not adequately examined trends in early detection or the adoption of emerging therapies, such as immunotherapy and targeted treatments, in real-world settings [11]. Additionally, most existing studies are limited to Western contexts, overlooking the unique challenges and opportunities in rapidly aging Asian societies [12]. To address these gaps, we conducted a 15-year cohort study to analyze trends and survival outcomes by cancer type and stage in patients aged 80 years and older. This study aims to descriptively evaluate temporal trends in cancer diagnosis, treatment patterns, and survival outcomes in patients aged 80 years and older using a 15-year hospital-based cancer registry. These findings will contribute to a better understanding of how cancer care is changing in the population of very old adults and could inform future guidelines and policies that support more individualized treatment decisions in rapidly aging populations over chronological age.

## 2. Methods

### 2.1. Study Population

This retrospective cohort study utilized the Samsung Medical Center Cancer Registry, which includes patients diagnosed with cancer between 2008 and 2022. Clinical information for these patients is routinely updated by a trained cancer data manager using the electronic medical record (EMR) system. The Institutional Review Board of Samsung Medical Center approved this study (SMC-2021-12-036) and waived the requirement for informed consent, as only de-identified data routinely collected during health screening visits were used. Additionally, the Samsung Medical Center Cancer Registry has been registered in clinicaltrial.gov (NCT06703957). The analysis was conducted using de-identified data on a secure hospital server, with access restricted to authorized personnel in accordance with institutional data protection policies.

### 2.2. Variables and Data Collection

All medical, administrative, and patient information is stored in an EMR system [13]. Age at diagnosis (years) was obtained from the EMR and categorized as under 80 or 80 years and older. Other sociodemographic and clinical characteristics, including gender, body mass index, region of residence, marital status at diagnosis, employment status at diagnosis, diagnostic pathways (screening-detected, incidentally diagnosed, symptom-detected), and diagnostic method, were extracted from the EMR by trained cancer registrars.

Cancer types were classified using the GLOBOCAN cancer dictionary [14] and the taxonomy adopted in Cancer Incidence in Five Continents [11], both of which were provided by the International Association of Cancer Registries. In this study, cancer classification with 24 cancer types was used, based on the Korea National Cancer Registry annual report [15]. The summary staging system developed under the Surveillance, Epidemiology, and End Results (SEER) program (i.e., SEER summary staging) [12] was used to categorize the extent of tumor invasion or metastasis. Based on the 7th edition of the American Joint Committee on Cancer Staging Manual, SEER data were enriched with tumor grades, invasion/metastasis status, site-specific variables, and tumor stages.

Treatment modalities included surgery, chemotherapy (cytotoxic chemotherapy, targeted therapy, immunotherapy), and radiation therapy, all of which were extracted from the DARWIN-C clinical data warehouse at Samsung Medical Center.

Survival status and date of death until May 2024 were obtained from the mortality database of the Ministry of the Interior and Safety. Overall survival was defined as the time from diagnosis to death from any cause.

### 2.3. Statistical Analysis

Baseline characteristics were compared across time periods using trend analysis. Kaplan–Meier survival analysis was conducted to estimate 5-year survival. Patients were followed from the date of cancer diagnosis until death, five years post-diagnosis, or the administrative end date (31 May 2024).

To compare age groups, multivariable Cox proportional hazards models were used to calculate hazard ratios (HRs) and 95% confidence intervals (CIs) for mortality. Mortality differences based on treatment were also assessed using multivariable Cox regression. Additionally, the interaction between age and treatment was examined to evaluate whether the effect of treatment on mortality varied by age. To evaluate whether the sample size of patients aged ≥80 years was sufficient for comparative survival analysis, we conducted a simplified power calculation. Assuming a 5-year survival rate of 50% in the treated group and 40% in the untreated group, with a treatment proportion of 30% and a significance level of 0.05, the minimum total sample size required to achieve 80% power was approximately 1000.

All analyses were two-sided, with *p*-values < 0.05 considered statistically significant. All statistical analyses were conducted using SAS version 9.4 (SAS Institute Inc., Cary, NC, USA) and R version 4.0.3 (R Foundation for Statistical Computing, Vienna, Austria).

## 3. Results

### 3.1. Trends in Elderly Patients and Characteristics

A total of 301,055 patients with cancer were included in this study, of whom 13,111 (4.4%) were aged 80 years or older at the time of diagnosis (Table 1). The proportion of patients aged 80 years or older steadily increased, rising from 2.4% in 2008 to 5.8% in 2022 (Figure 1A). Notably, among all cancer types, bladder cancer had the highest proportion of patients aged ≥80 years (Appendix A).

The mean age at diagnosis for patients aged ≥80 years was 83.1 years. Older patients were more likely to be male (60.4% vs. 52.7%, *p* < 0.01), underweight (6.6% vs. 4.1%, *p* < 0.01), and unmarried (16.0% vs. 10.8%, *p* < 0.01). Among patients aged ≥80 years, 39.3% had their cancer detected due to symptoms, a higher proportion than in younger patients. Regarding cancer type distribution, patients aged ≥80 years had higher proportions of lung, prostate, and bladder cancers compared to younger patients, while thyroid and breast cancers were less common in the older group. The most frequently diagnosed cancers in patients aged ≥80 years were lung (18.9%), stomach (15.3%), and colorectal cancer (13.8%). Patients in this age group were more likely to receive an initial diagnosis before transferring to another hospital and were less likely to undergo all types of treatment than younger patients (Table 1).

### 3.2. Survival Outcomes by Age and Stage

Overall, patients aged ≥80 years had lower survival rates than younger patients, with 5-year survival rates of 38.42% versus 70.39% for all cancer stages combined (HR = 1.30, 95% CI = 1.26, 1.33; Table 2). Notably, among patients aged ≥80 years, those with breast cancer (HR = 3.23; 95% CI = 2.40, 4.36) and colorectal cancer (HR = 2.07; 95% CI = 1.90, 2.26) had significantly lower survival rates than younger patients (Table 2). Among individuals with localized or regional-stage cancers, the 5-year survival rate was 49.66% for those aged ≥80 years, compared to 81.46% for younger patients (HR = 1.41; 95% CI = 1.35, 1.46; Table 2, Figure 2I). For distant-stage cancers, survival was lower, at 10.53% for patients aged ≥80 years versus 27.61% for those aged <80 years (HR = 1.14; 95% CI = 1.10, 1.19; Table 2, Figure 2II).

### 3.3. Effect of Treatment on Survival

Among patients aged 80 years and older, 55% received anti-cancer treatment, with the proportion increasing from 54.5% in 2008 to 60.3% in 2021 (Figure 1B). Notably, this increase was particularly pronounced in patients with distant-stage disease (Figure 1C). Additionally, there was a rising trend in the proportion of clinical trial participants aged ≥80 years (Figure 1D). Patients who received anti-cancer treatment were more likely to be younger, female, and have localized cancer (Appendix A).

Patients aged ≥80 years had lower survival rates than younger patients among those who received treatment (HR = 1.22; 95% CI = 1.17, 1.27; Figure 3, Table 2). In subgroup analysis by stage, the HR for mortality in patients aged ≥80 years was 1.38 (95% CI = 1.31, 1.46) for localized or regional disease and 1.01 (95% CI = 0.95, 1.08) for distant disease (Table 2).

Regarding treatment effect, patients aged ≥80 years who received treatment had significantly better survival outcomes than those who did not receive treatment (HR = 0.45; 95% CI = 0.42, 0.49), comparable to younger patients (HR = 0.42; 95% CI = 0.41, 0.43). These findings were consistent across both localized or regional disease (HR = 0.45; 95% CI = 0.42, 0.49) and distant-stage disease (HR = 0.58; 95% CI = 0.53, 0.62) (Figure 4).

## 4. Discussion

This comprehensive cohort study from the SMC Cancer Registry at a major Korean medical center revealed several key findings. First, the number of elderly patients aged ≥80 years increased. Second, the proportion of treated patients also rose, particularly after 2020, when more than 60% of patients aged ≥80 years received treatment. Third, although mortality was higher in this age group than in younger patients, those aged ≥80 years who received treatment experienced survival benefits comparable to younger patients across all cancer types.

The rising number of elderly patients with cancer aligns with global demographic shifts toward aging populations. By 2050, an estimated 6.9 million new cancer cases will be diagnosed annually in adults aged ≥80 years worldwide, accounting for 20.5% of all cancer cases [16]. In the United States, adults aged ≥85 years (the “oldest old”) represent the fastest-growing age group [1]. Notably, “super-aged” societies such as South Korea, Japan, and China have reported a growing proportion of elderly individuals diagnosed with cancer [12,17,18]. While these trends reflect global patterns, the increasing number of older patients with cancer also suggests improved survival rates and enhanced detection methods [19]. Healthcare systems must prepare for a growing elderly cancer population, emphasizing resource allocation for geriatric oncology services and specialized training for healthcare professionals in elderly care [20].

Previous studies have reported that treatment rates in this age group rarely exceeded 30%, primarily due to concerns about frailty and potential toxicity [4]. However, our findings indicate a significant rise in treatment rates, with over 60% of patients aged ≥80 years receiving treatment. This shift may be attributed to the introduction of more tolerable therapies, such as immune checkpoint inhibitors and targeted treatments, which reduce toxicity while maintaining therapeutic efficacy [8]. Interestingly, our analysis showed that the proportion of treated patients aged ≥80 years with distant-stage disease has increased since 2018. In South Korea, oncologists have been able to use immune checkpoint inhibitors for patients with advanced-stage cancer in routine clinical practice. Our data likely reflect the medical landscape during this period. Additionally, improved access to healthcare infrastructure may have contributed to this rise [21]. The increased treatment rates among elderly patients underscore the need for the continuous development of supportive care systems to mitigate potential side effects in older populations [11]. The incorporation of geriatric assessments into oncology practices is essential for clinicians to better evaluate the functional reserve of elderly patients, enabling more tailored and feasible treatment options.

In this study, the survival rate among patients aged ≥80 years was lower than that of younger patients, even after receiving treatment. These results are consistent with previous studies [22,23]. Aging is associated with diminished organ function and a reduced ability to recover from the physical stress of cancer treatments such as surgery, chemotherapy, or radiation therapy [21,24]. This reduced physiological reserve makes older patients more vulnerable to treatment-related side effects and complications, potentially increasing mortality [21]. However, our findings indicate that patients aged ≥80 years who received treatment experienced survival benefits comparable to younger patients across all cancer types. This represents a significant advancement in our understanding of treatment efficacy in this age group and challenges previous assumptions about the futility of aggressive treatments for the elderly. Notably, our analysis also demonstrated that the proportion of clinical trial participants aged ≥80 years has been increasing since 2015. Generally, most clinical trials have inclusion criteria that limit age, and oncologists are often reluctant to enroll elderly patients in trials involving new investigational drugs. Since 2015, our institution has operated a Personalized Cancer Clinic (currently named the Precision Cancer Center), which includes highly specialized oncologists, clinical research coordinators, nurses, and cancer researchers. This specialized organization may have played a crucial role in increasing the proportion of elderly patients with cancer participating in clinical trials, as well as expanding access to systemic anti-cancer treatment.

### 4.1. Perspectives for Clinical Practice

Based on our results, there is a clear need for geriatric-specific oncology services that assess not only chronological age but also physiological reserves and functional status. The observed survival benefit associated with active treatment, even in this advanced age group, underscores that age alone should not be a barrier to therapeutic intervention [5]. Moreover, incorporating geriatric assessments into routine oncology care may enable more personalized and appropriate treatment planning, ensuring that functionally fit older patients are not excluded from potentially curative or disease-controlling therapies [25]. Finally, the upward trend in clinical trial participation among older adults in our cohort suggests a shifting landscape in which this population is increasingly being considered for investigational therapies. Expanding trial eligibility criteria and promoting inclusive trial designs will be critical for generating evidence that more accurately reflects the real-world aging population. Further research and guideline development are needed to support more precise, individualized cancer care for older adults.

### 4.2. Limitations

This study had several limitations. First, although treatment rates and outcomes were evaluated, this study did not extensively analyze the reasons for non-treatment in the elderly population. Second, this study primarily focused on survival outcomes and did not assess quality of life or functional status, which are critical considerations in geriatric oncology. Future studies should incorporate these measures to provide a more comprehensive understanding of treatment effects in older adults. Third, rare cancers were not analyzed separately due to small sample sizes. Nevertheless, compared to previous studies, our cohort included a substantially larger number of older patients with cancer, which enabled meaningful survival comparisons by treatment status even within this advanced age group. Fourth, this analysis had selection bias. The decision to administer anti-cancer treatment in the elderly population significantly impacts both quality of life and survival outcomes. It is possible that elderly patients who did not receive anti-cancer treatment in this analysis were ineligible due to poor performance status, insufficient organ function, or severe comorbidities. Therefore, these findings must be interpreted with caution. Lastly, as the study was conducted at a single tertiary referral center in South Korea, the generalizability of our findings may be limited. Patient characteristics, institutional infrastructure, and care delivery models can differ substantially across hospitals and countries, particularly in community or rural settings.

## 5. Conclusions

In conclusion, the number of elderly patients aged ≥80 years has been increasing, along with their treatment rates. Considering that patients aged ≥80 years who received treatment experienced survival benefits comparable to younger patients across all cancer types, further guideline development is essential to optimize geriatric oncology care.

## Figures and Tables

**Figure 1 cancers-17-02017-f001:**
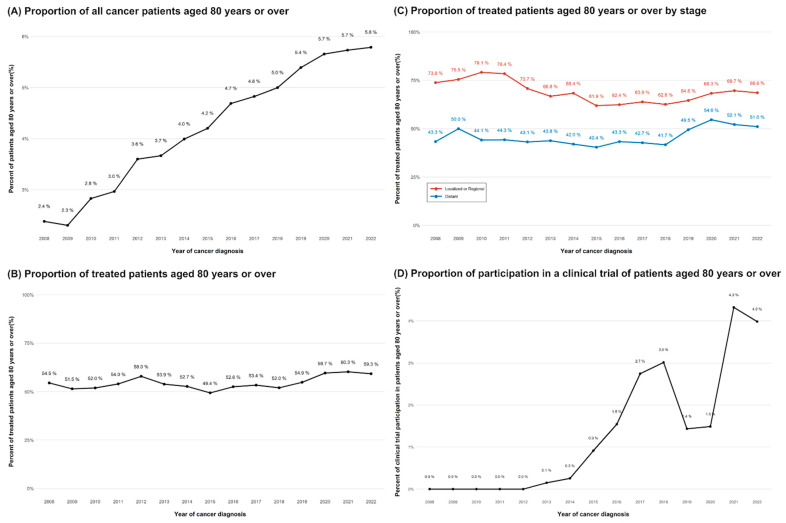
Trends in the proportion of patients aged 80 years or more receiving treatment (2008–2022).

**Figure 2 cancers-17-02017-f002:**
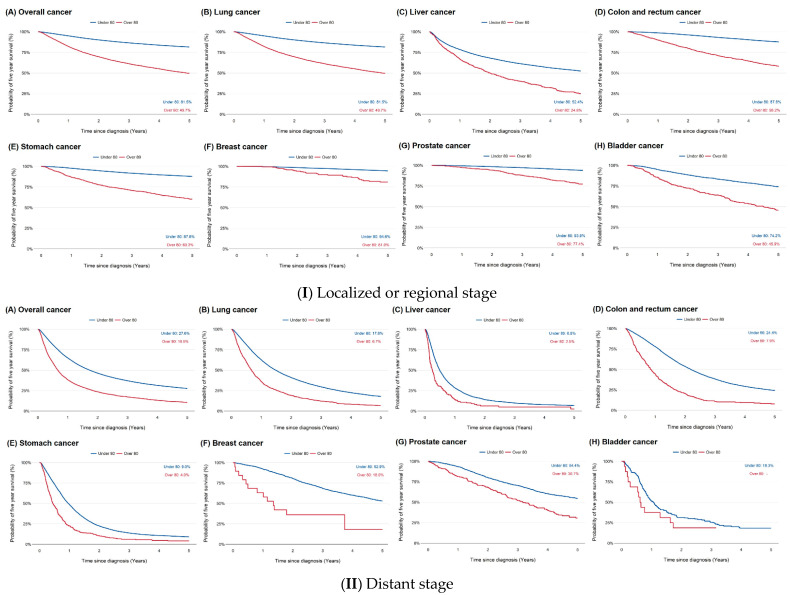
Kaplan–Meier survival curves for patients aged <80 vs. ≥80 years.

**Figure 3 cancers-17-02017-f003:**
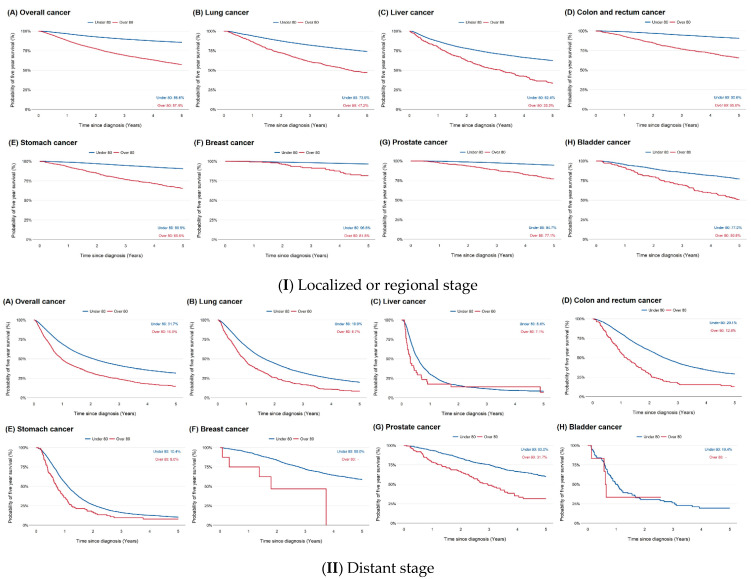
Kaplan–Meier survival curves for treated vs. untreated patients aged ≥80 years.

**Figure 4 cancers-17-02017-f004:**
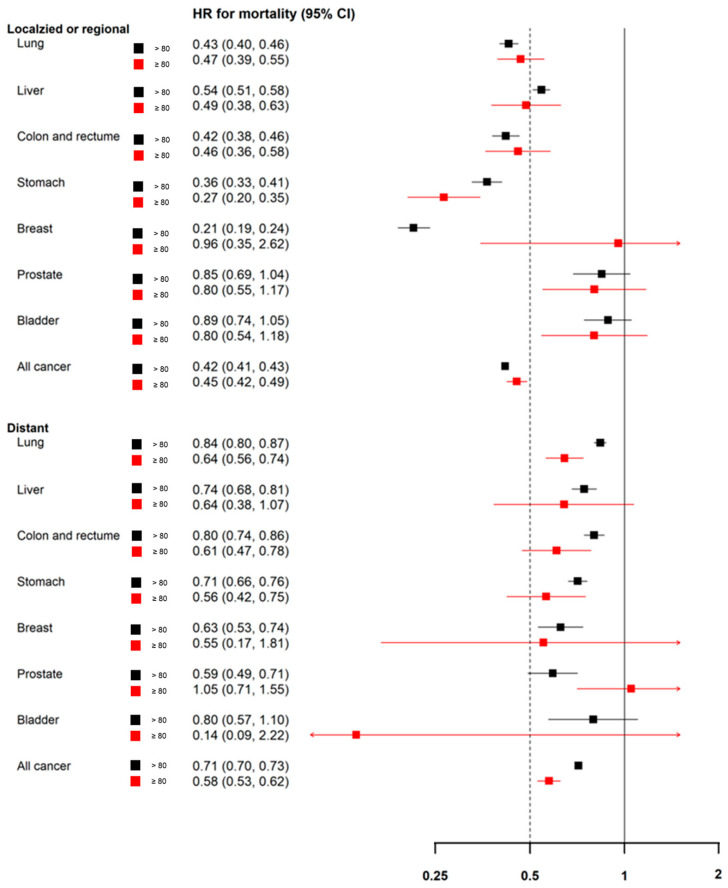
Effect of treatment on mortality in older patients with cancer.

**Table 1 cancers-17-02017-t001:** Baseline characteristics.

Characteristics	<80	≥80	*p* Value
	N = 287,944	N = 13,111	
**Age at diagnosis (years), mean (SD)**	56.6 (13.6)	83.1 (3.1)	<0.01
**Gender, male**	151,661 (52.7%)	7918 (60.4%)	<0.01
**Body mass index (kg/m^2^)**			
Underweight (≤18.5 kg/m^2^)	11,726 (4%)	868 (6%)	
Normal (18.5–23 kg/m^2^)	96,323 (33.5%)	4297 (32.8%)	
Overweight (23–25 kg/m^2^)	60,939 (21.2%)	2541 (19.4%)	
Obese (>25 kg/m^2^)	82,480 (28.6%)	2825 (21.5%)	
Unknown	36,476 (12.7)	2580 (19.7)	
**Residence area**			<0.01
Seoul	56,444 (19.6%)	3671 (28.0%)	
Others	221,543 (76.9%)	8926 (68.1%)	
Unknown	9957 (3.5%)	514 (3.9%)	
**Marital status at diagnosis, married**			<0.01
Uncoupled (unmarred, separated, widowed)	31,171 (10.8%)	2097 (16.0%)	
Coupled	193,356 (67.2%)	6865 (52.4%)	
Unknown	63,417 (22.0%)	4149 (31.6%)	
**Working status at diagnosis**			<0.01
No work	187,078 (65.0)	11,389 (86.9%)	
White color	60,708 (21.1%)	590 (4.5%)	
Blue color	22,755 (7.9%)	831 (6.3%)	
Service	4468 (1.6%)	34 (0%)	
Others	795 (0%)	21 (0%)	
Unknown	12,140 (4.2%)	246 (2%)	
**Diagnosis path (2013–)**			<0.01
Detected from Health screening	102,627 (35.6%)	4023 (30.7%)	
Incidentally diagnosed cancer	7304 (2.5%)	619 (4.7%)	
Detected due to symptoms	75,875 (26.4%)	5152 (39.3%)	
Unknown	19,880 (6.9%)	950 (7.2%)	
**Type of cancer**			<0.01
Lip, oral cavity and pharynx	5262 (1.8%)	199 (1.5%)	
Esophagus	5897 (2.0%)	297 (2.3%)	
Stomach	37,580 (13.1%)	2011 (15.3%)	
Colon and rectum	29,729 (10.3%)	1813 (13.8%)	
Liver	22,083 (7.7%)	798 (6.1%)	
Gallbladder, etc.	5817 (2.0%)	628 (4.8%)	
Pancreas	7797 (2.7%)	640 (4.9%)	
Larynx	1093 (0.4%)	77 (0.6%)	
Lung	38,308 (13.3%)	2483 (18.9%)	
Breast	32,916 (11.4%)	244 (1.9%)	
Cervix uteri	4517 (1.6%)	108 (0.8%)	
Corpus uteri	3713 (1.3%)	48 (0.4%)	
Ovary	4097 (1.4%)	66 (0.5%)	
Prostate	14,045 (4.9%)	1056 (8.1%)	
Testis	414 (0.0%)	1 (0.0%)	
Kidney	7951 (2.8%)	217 (1.7%)	
Bladder	3489 (1.2%)	428 (3.3%)	
Brain and central nervous system (CNS)	4226 (1.5%)	121 (0.9%)	
Thyroid	28,397 (9.9%)	131 (1.0%)	
Hodgkin lymphoma	530 (0.2%)	7 (0.1%)	
Non-Hodgkin lymphoma	7477 (2.6%)	325 (2.5%)	
Multiple myeloma	1799 (0.6%)	104 (0.8%)	
Leukemia	2979 (1.0%)	85 (0.6%)	
Other and unspecified	17,828 (6.2%)	1224 (9.3%)	
**SEER stage**			<0.01
Localized	113,145 (39.3%)	4169 (31.8%)	
Regional	94,987 (33%)	4199 (32.0%)	
Distant	53,587 (18.6%)	3154 (24.1%)	
Unknown	26,225 (9.1%)	1589 (12.1%)	
**Diagnosis and treatment**			<0.01
SMC diagnosis, first treatment at SMC	81,571 (28.3%)	4032 (30.8%)	
Diagnosis elsewhere, first treatment at SMC	115,692 (40.2%)	3289 (25.1%)	
SMC diagnosis, first treatment elsewhere	397 (0.1%)	22 (0.2%)	
Diagnosis and first treatment elsewhere	48,591 (16.9%)	1751 (13.4%)	
SMC diagnosis only	41,693 (14.5%)	4017 (30.6%)	
**Treatment within 4 months (N = 203,181)**			
Surgery	141,272 (72.1%)	4247 (58.6%)	<0.01
Chemotherapy (cytotoxic, targeted, immunotherapy)	79,746 (40.7%)	1850 (25.5%)	<0.01
Radiotherapy	31,895 (16.3%)	1238 (17.1%)	0.07
Hormone therapy	13,421 (6.8%)	513 (7.1%)	0.46
Biochemical therapy	3245 (1.7%)	164 (2.3%)	<0.01
Others	12,203 (6.2%)	553 (7.6%)	<0.01

**Table 2 cancers-17-02017-t002:** Five-year observed survival rate by age group.

	<80	≥80	<80 vs. ≥80 HR (95% CI)
**Overall**			
**Any stages**			
**All cancer**	70.39 (70.22%, 70.57%)	38.42 (37.53%, 39.53%)	1.30 (1.26, 1.33)
Lung cancer	47.03 (46.50%, 47.56%)	24.07 (22.28%, 26.01%)	1.22 (1.15, 1.29)
Liver cancer	45.36 (44.68%, 46.04%)	22.20 (19.25%, 25.60%)	1.63 (1.48, 1.79)
Colon and rectum cancer	73.68 (73.15%, 74.21%)	45.81 (43.42%, 48.34%)	2.07 (1.90, 2.26)
Stomach cancer	76.59 (76.15%, 77.03%)	47.53 (45.26%, 49.91%)	1.92 (1.77, 2.08)
Breast cancer	92.03 (91.72%, 92.35%)	73.61 (67.71%, 80.03%)	3.23 (2.40, 4.36)
Prostate cancer	89.27 (88.71%, 89.84%)	65.51 (62.28%, 68.91%)	1.56 (1.32, 1.85)
Bladder cancer	71.33 (69.76%, 72.96%)	46.32 (41.46%, 51.76%)	1.35 (1.11, 1.64)
**Localized or Regional**			
**All cancer**	81.46 (81.28%, 81.63%)	49.66 (48.50%, 50.85%)	1.41 (1.35, 1.46)
Lung cancer	68.55 (67.89%, 69.22%)	38.65 (35.72%, 41.81%)	1.31 (1.20, 1.44)
Liver cancer	52.40 (51.61%, 53.20%)	24.77 (21.11%, 29.07%)	1.61 (1.44, 1.80)
Colon and rectum cancer	87.64 (87.17%, 88.11%)	58.22 (55.37%, 61.22%)	2.10 (1.85, 2.38)
Stomach cancer	87.81 (87.43%, 88.20%)	60.34 (57.61%, 63.20%)	1.76 (1.56, 1.98)
Breast cancer	94.58 (94.30%, 94.87%)	80.98 (74.94%, 87.50%)	2.90 (1.94, 4.34)
Prostate cancer	93.93 (93.44%, 94.42%)	77.37 (73.86%, 81.06%)	1.24 (0.96, 1.60)
Bladder cancer	74.20 (72.55%, 75.90%)	45.91 (40.65%, 51.86%)	1.31 (1.06, 1.63)
**Distant**			
**All cancer**	27.61 (27.21%, 28.02%)	10.53 (9.34%, 11.87%)	1.14 (1.10, 1.19)
Lung cancer	17.82 (17.16%, 18.50%)	6.66 (5.08%, 8.74%)	1.17 (1.08, 1.26)
Liver cancer	6.50 (5.59%, 7.57%)	2.47 (0.49%, 12.49%)	1.61 (1.31, 1.97)
Colon and rectum cancer	24.41 (23.28%, 25.61%)	7.90 (5.34%, 11.69%)	1.83 (1.61, 2.08)
Stomach cancer	9.04 (8.19%, 9.97%)	4.00 (2.11%, 7.57%)	1.62 (1.40, 1.87)
Breast cancer	52.94 (50.38%, 55.63%)	18.05 (3.97%, 81.96%)	2.94 (1.61, 5.36)
Prostate cancer	54.42 (51.64%, 57.34%)	30.73 (24.11%, 39.16%)	1.61 (1.26, 2.07)
Bladder cancer	18.30 (13.07%, 25.64%)	-	1.36 (0.68, 2.71)
**Treated**			
**Any stages**			
**All cancer**	76.54 (76.34%, 76.73%)	48.53 (47.27%, 49.81%)	1.22 (1.17, 1.27)
Lung cancer	54.35 (53.72%, 54.97%)	32.14 (29.43%, 35.09%)	1.14 (1.05, 1.24)
Liver cancer	56.09 (55.21%, 56.99%)	30.99 (26.11%, 36.79%)	1.61 (1.40, 1.86)
Colon and rectum cancer	79.76 (79.17%, 80.35%)	58.20 (55.21%, 61.36%)	2.09 (1.85, 2.35)
Stomach cancer	83.44 (82.99%, 83.88%)	65.67 (62.71%, 68.78%)	1.82 (1.60, 2.07)
Breast cancer	94.82 (94.63%, 95.21%)	78.54 (71.85%, 85.86%)	2.89 (1.91, 4.38)
Prostate cancer	91.59 (90.96%, 92.22%)	66.78 (62.31%, 71.56%)	1.75 (1.37, 2.22)
Bladder cancer	74.20 (72.05%, 76.41%)	50.67 (43.90%, 58.48%)	1.19 (0.89, 1.59)
**Localized or Regional**			
**All cancer**	85.62 (85.44%, 85.88%)	57.36 (55.95%, 58.81%)	1.38 (1.31, 1.46)
Lung cancer	73.89 (73.19%, 74.59%)	47.20 (43.40%, 51.34%)	1.27 (1.12, 1.44)
Liver cancer	62.36 (61.43%, 63.30%)	33.29 (28.05%, 39.52%)	1.62 (1.38, 1.89)
Colon and rectum cancer	90.60 (90.12%, 91.08%)	65.57 (62.42%, 68.88%)	2.03 (1.72, 2.39)
Stomach cancer	90.71 (90.34%, 91.08%)	71.43 (68.42%, 74.58%)	1.64 (1.40, 1.92)
Breast cancer	96.58 (96.34%, 96.83%)	81.79 (75.17%, 88.99%)	3.32 (2.06, 5.36)
Prostate cancer	94.66 (94.11%, 95.20%)	77.08 (72.43%, 82.03%)	1.27 (0.91, 1.78)
Bladder cancer	76.99 (74.84%, 79.20%)	50.59 (43.69%, 58.58%)	1.14 (0.84, 1.54)
**Distant**			
**All cancer**	31.73 (31.19%, 32.28%)	14.99 (12.93%, 17.36%)	1.01 (0.95, 1.08)
Lung cancer	19.92 (19.07%, 20.81%)	8.71 (6.19%, 12.25%)	1.08 (0.97, 1.20)
Liver cancer	8.42 (6.99%, 10.14%)	7.06 (1.39%, 35.84%)	1.43 (0.98, 2.10)
Colon and rectum cancer	29.06 (27.47%, 30.73%)	12.82 (8.10%, 20.29%)	1.65 (1.35, 2.01)
Stomach cancer	10.44 (9.21%, 11.83%)	7.97 (3.81%, 16.67%)	1.59 (1.25, 2.00
Breast cancer	59.00 (59.95%, 62.22%)	-	2.07 (0.81, 5.31)
Prostate cancer	60.02 (56.27%, 64.02%)	31.68 (23.05%, 43.55%)	2.00 (1.42, 2.81)
Bladder cancer	19.38 (12.21%, 30.76%)	-	1.45 (0.53, 3.99)

Adjusted for gender, marital status, job, body mass index, and SEER stage.

## Data Availability

The data supporting the findings of this study are available from the corresponding author upon reasonable request.

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
