# Peer review of "Changes in Cancer Care for Patients Aged 80 and Above: A Cohort Study from Samsung Comprehensive Cancer Center in South Korea"

_cancers, 2025, doi:10.3390/cancers17122017_

Round 1

Reviewer 1 Report

Comments and Suggestions for Authors

Dear Authors,

First of all, I would like to express my sincere thanks for the opportunity to contribute my opinion to the evaluation of your manuscript. I found the topic extremely interesting and highly relevant to our field. Below, I outline the main areas that could benefit from further elaboration and revision.

Editing
Please pay attention to the formatting of the references according to the journal’s editorial guidelines. Additionally, I recommend replacing "sex" with "gender" throughout the manuscript and ensuring that all tables include appropriate legends for the acronyms used.

Title
The population under study is missing (e.g., "Germany").

Abstract
Well written. However, I suggest placing greater emphasis on potential implications for clinical practice (see following comments) and restructuring it using the standard format: Introduction/Objectives, Methods, Results, and Conclusions.

Keywords
Currently missing.

Introduction
The introduction should be revised and better structured in terms of content. I believe that presenting healthy lifestyle and nutrition as comprehensive approaches for chronic disease management, and then transitioning directly to cancer, is not sufficiently coherent. For these reasons, I suggest expanding this section with recent and appropriate literature on the following topics: “Mediterranean Diet and Lifestyle Medicine for Support and Care of Patients with Type II Diabetes”, “The Anti-Inflammatory Effects of a Mediterranean Diet”, and “Approach to Obesity Treatment in Primary Care” to then expand the discussion of this part to cancer, including potential topics such as “Diet and Lifestyle in Cancer Prevention”, “Quality of Life in Women with Breast Cancer after Treatment of Lifestyle Modifications”, and “Diet and Carcinogenesis of Gastric Cancer”; these additions would significantly strengthen this critical section of the manuscript
and broaden its potential readership and dissemination. The objectives are somewhat unclear. I recommend using the standard format of clearly stating the primary objectives and then the secondary objectives, e.g., “The primary objectives of the study were... whereas the secondary objectives were...”. If possible, also include the research questions the authors aim to answer.

Methods
This section lacks clarity in several areas, particularly concerning ethics and data protection.

Results
This is undoubtedly the strongest section of the manuscript, but it could further benefit from the suggestions mentioned above and those that follow.

Discussion
In line with the introduction, this section should be supported by stronger evidence of effectiveness, which is currently lacking. I also recommend adapting the discussion to a clinical practice perspective, possibly by including a dedicated section such as “Perspectives for Clinical Practice” in which to discuss the potential applications of the findings in real- world settings, ideally from a multidisciplinary standpoint.

Limitations
I suggest creating a dedicated section for limitations, focusing particularly on the generalizability (or lack thereof) of the findings.

Conclusions
Should be revised according to the above suggestions and presented in a separate section.

References
The current bibliography does not adequately support the manuscript. It should be expanded as per the suggestions provided. I also recommend updating any references older than 10 years, unless they are of methodological relevance or provide high levels of evidence.

In summary, the manuscript presents scientifically interesting results. With the proposed improvements, it could significantly increase its overall quality. My recommendation is to proceed with a thorough revision as suggested, as the manuscript—if appropriately modified—could represent a meaningful contribution to the relevant scientific literature.

Author Response

Response to Reviewer #1

General comments

First of all, I would like to express my sincere thanks for the opportunity to contribute my opinion to the evaluation of your manuscript. I found the topic extremely interesting and highly relevant to our field. Below, I outline the main areas that could benefit from further elaboration and revision.

Response: We thank the reviewer for the thorough and insightful review. We have carefully addressed each of the specific comments and revised the manuscript to improve.

Specific comments

  1. Editing

Please pay attention to the formatting of the references according to the journal’s editorial guidelines. Additionally, I recommend replacing "sex" with "gender" throughout the manuscript and ensuring that all tables include appropriate legends for the acronyms used.

Response: We have revised all reference formatting in accordance with the journal’s editorial guidelines. We also replaced the term “sex” with “gender” throughout the manuscript and carefully reviewed all tables to ensure that acronyms are fully defined in the legends.

  1. Title

The population under study is missing (e.g., "Germany").

Response: In response to the reviewer’s comment, we have revised the Title included population under study as follows

“Changes in Cancer Care for Patients Aged 80 and Above: A Cohort Study from Samsung Comprehensive Cancer Center in South Korea”

  1. Abstract

Well written. However, I suggest placing greater emphasis on potential implications for clinical practice (see following comments) and restructuring it using the standard format: Introduction/Objectives, Methods, Results, and Conclusions.

Response: In response to the reviewer’s comment, we have revised the abstract using the standard format as follows

“ABSTRACT

Background/Objectives: With an estimated 70% of new cancer diagnoses expected in older adults within the next decade, cancer care for this population has gained increasing global attention. Additionally, older patients are less likely to receive optimal cancer treatments.

Methods: This retrospective cohort study utilized data from the Samsung Medical Center Cancer Registry, which includes patients diagnosed with cancer between 2008 and 2022. A 15-year cohort analysis was conducted to examine trends and survival outcomes by cancer type and stage in patients aged 80 years and older.

Results: Among 301,055 patients with cancer, 13,111 (4.4%) were aged 80 years or older at diagnosis. The proportion of patients in this age group increased from 2.4% in 2008 to 5.8% in 2022. The most prevalent cancers in patients aged ≥80 years were lung (18.9%), stomach (15.3%), and colorectal cancer (13.8%). Among individuals with localized or regional-stage disease, the 5-year survival rate was 49.66% in those aged ≥80 years compared to 81.46% in younger patients (HR = 1.41; 95% CI = 1.35, 1.46). For distant-stage disease, survival was lower at 10.53% in patients aged ≥80 years versus 27.61% in those aged <80 (HR = 1.14; 95% CI = 1.10, 1.19). Among patients aged 80 years and older, 55% received anti-cancer treatment, with the proportion increasing from 54.5% in 2008 to 60.3% in 2021. This increase was particularly notable in individuals with distant-stage disease. Additionally, the proportion of clinical trial participants aged ≥80 years exhibited an upward trend. Patients in this age group who underwent treatment had significantly improved survival compared to those who did not, in both localized or regional disease (HR = 0.45; 95% CI = 0.42, 0.49) and distant disease (HR = 0.58; 95% CI = 0.53, 0.62).

Conclusion: Findings from this cohort of the SMC Cancer Registry highlight key trends, including a rising number of patients aged ≥80 years and an increasing proportion receiving treatment, particularly after 2020, when more than 60% received therapy. Furthermore, survival benefits associated with treatment were comparable to those observed in younger patients across all cancer types."

  1. Keywords

Currently missing.

Response: In response to the reviewer’s comment, we have added the following keywords:

“Key words: elderly patients, geriatric oncology, survival, treatment trends, real-world data”

  1. Introduction

The introduction should be revised and better structured in terms of content. I believe that presenting healthy lifestyle and nutrition as comprehensive approaches for chronic disease management, and then transitioning directly to cancer, is not sufficiently coherent. For these reasons, I suggest expanding this section with recent and appropriate literature on the following topics: “Mediterranean Diet and Lifestyle Medicine for Support and Care of Patients with Type II Diabetes”, “The Anti-Inflammatory Effects of a Mediterranean Diet”, and “Approach to Obesity Treatment in Primary Care” to then expand the discussion of this part to cancer, including potential topics such as “Diet and Lifestyle in Cancer Prevention”, “Quality of Life in Women with Breast Cancer after Treatment of Lifestyle Modifications”, and “Diet and Carcinogenesis of Gastric Cancer”; these additions would significantly strengthen this critical section of the manuscript

and broaden its potential readership and dissemination. The objectives are somewhat unclear. I recommend using the standard format of clearly stating the primary objectives and then the secondary objectives, e.g., “The primary objectives of the study were... whereas the secondary objectives were...”. If possible, also include the research questions the authors aim to answer.

Response: We thank the reviewer for this valuable and constructive comment. In response, we have revised the Introduction to better focus on the purpose of the study. In light of increasing cancer diagnoses among healthier older adults due to general improvements in overall health, nutrition, and health-related behaviors, we now emphasize that treatment decisions should be guided by frailty and functional status rather than chronological age. This perspective is supported by recent literature demonstrating the feasibility and benefit of active treatment in older patients who are physiologically fit. Furthermore, we have revised the objective statement to clearly articulate the primary aim of the study, which was to descriptively evaluate trends in cancer diagnosis, treatment patterns, and survival outcomes among patients aged 80 years and older. These revisions are reflected in the following sections of the revised Introduction.

Introduction (page 2, line 59-65, and line 77-83)

“A significant proportion of older adults are less likely to receive optimal cancer treatments. According to a prior study, only 30% of patients aged 80 years and older received chemotherapy, compared to 65% of those aged 18-59 years​.[4] However, recent studies suggest that older adults in good functional health can still derive meaningful survival benefits from active cancer treatment, particularly when treatment decisions are based on frailty rather than chronological age.[5] In fact, recent improvements in general health, driven by better nutrition, healthier lifestyles, [6] and expanded access to cancer screening, have contributed to earlier detection. This has led to a growing subset of older patients who are able to undergo and benefit from active treatment.[7,8]

“To address these gaps, we conducted a 15-year cohort study to analyze trends and survival outcomes by cancer type and stage in patients aged 80 years and older. This study aims to descriptively evaluate temporal trends in cancer diagnosis, treatment patterns, and survival outcomes in patients aged 80 years and older using a 15-year hospital-based cancer registry. These findings will contribute to a better understanding of how cancer care is changing in the population of very old adults, and could inform future guidelines and policies that support more individualized treatment decisions in rapidly aging populations over chronological age.”

  1. Methods

This section lacks clarity in several areas, particularly concerning ethics and data protection.

We appreciate the reviewer’s concern regarding ethics and data protection. To address this, we have clarified in the revised Methods section that all data were fully de-identified prior to analysis and accessed only by authorized personnel through secure institutional systems, in accordance with established data protection protocols. These clarifications have been added to the “Study Population” subsection of the Methods.

Methods (page 2, line 94-96)

“The Institutional Review Board of Samsung Medical Center approved this study (SMC-2021-12-036) and waived the requirement for informed consent, as only de-identified data routinely collected during health screening visits were used. Additionally, the Samsung Medical Center Cancer Registry has been registered in clinicaltrial.gov (NCT06703957). The analysis was conducted using de-identified data on a secure hospital server, with access restricted to authorized personnel in accordance with institutional data protection policies.

  1. Results

This is undoubtedly the strongest section of the manuscript, but it could further benefit from the suggestions mentioned above and those that follow.

Response:
We appreciate the positive evaluation.

  1. Discussion

In line with the introduction, this section should be supported by stronger evidence of effectiveness, which is currently lacking. I also recommend adapting the discussion to a clinical practice perspective, possibly by including a dedicated section such as “Perspectives for Clinical Practice” in which to discuss the potential applications of the findings in real- world settings, ideally from a multidisciplinary standpoint.

Response: We thank the reviewer for this insightful suggestion. As recommended, we have adapted the structure of the Discussion section to better highlight clinical implications. Specifically, we added a new subsection titled “Perspectives for Clinical Practice” in which we emphasize the potential applications of our findings in real-world geriatric oncology practice. This includes the observed survival benefit of treatment in older adults, the need to base treatment decisions on frailty rather than chronological age, and the importance of expanding clinical trial access for the elderly.

Discussion (page 12, line 263-276)

“Perspectives for Clinical Practice

Based on our results, there is a clear need for geriatric-specific oncology services that assess not only chronological age, but also physiological reserve and functional status. The observed survival benefit associated with active treatment, even in this advanced age group, underscores that age alone should not be a barrier to therapeutic intervention. [5] Moreover, incorporating geriatric assessments into routine oncology care may enable more personalized and appropriate treatment planning, ensuring that functionally fit older patients are not excluded from potentially curative or disease-controlling therapies.[25] Finally, the upward trend in clinical trial participation among older adults in our cohort suggests a shifting landscape in which this population is increasingly being considered for investigational therapies. Expanding trial eligibility criteria and promoting inclusive trial designs will be critical for generating evidence that more accurately reflects the real-world aging population. Further research and guideline development are needed to support more precise, individualized cancer care for older adults.

  1. Limitations

I suggest creating a dedicated section for limitations, focusing particularly on the generalizability (or lack thereof) of the findings.

Response: We thank the reviewer for the helpful suggestion. In response, we have created a dedicated “Limitations” section in Discussion. We revised the text to more explicitly address the generalizability of our findings in the limitation section as follows.

Discussion (page 12, line 278-296)

“Limitation

This study had several limitations. First, although treatment rates and outcomes were evaluated, the study did not extensively analyze the reasons for non-treatment in the elderly population. Second, the study primarily focused on survival outcomes and did not assess quality of life or functional status, which are critical considerations in geriatric oncology. Future studies should incorporate these measures to provide a more comprehensive understanding of treatment effects in older adults. Limitation

This study had several limitations. First, although treatment rates and outcomes were evaluated, the study did not extensively analyze the reasons for non-treatment in the elderly population. Second, the study primarily focused on survival outcomes and did not assess quality of life or functional status, which are critical considerations in geriatric oncology. Future studies should incorporate these measures to provide a more comprehensive understanding of treatment effects in older adults. Third, rare cancers were not analyzed separately due to small sample sizes. Nevertheless, compared to previous studies, our cohort included a substantially larger number of older patients with cancer, which enabled meaningful survival comparisons by treatment status even within this advanced age group. Fourth, this analysis had selection bias. The decision to administer anti-cancer treatment in the elderly population significantly impacts both quality of life and survival outcomes. It is possible that elderly patients who did not receive anti-cancer treatment in this analysis were ineligible due to poor performance status, insufficient organ function, or severe comorbidities. Therefore, these findings must be interpreted with caution. Lastly, as the study was conducted at a single tertiary referral center in South Korea, the generalizability of our findings may be limited. Patient characteristics, institutional infrastructure, and care delivery models can differ substantially across hospitals and countries, particularly in community or rural settings.”

  1. Conclusions

Should be revised according to the above suggestions and presented in a separate section.

Response: In response, we have created a dedicated “Conclusions” section at the end of the Discussion.

  1. References

The current bibliography does not adequately support the manuscript. It should be expanded as per the suggestions provided. I also recommend updating any references older than 10 years, unless they are of methodological relevance or provide high levels of evidence.

Response: We appreciate the reviewer’s comment regarding the references. In response to earlier comments, we incorporated new references related to functional status, and geriatric oncology. With respect to the currency of the references, upon careful review, we note that the majority of references cited in the manuscript are from the past five to seven years, with the oldest dating to 2018.

Introduction (page 2, line 59-65, and line 77-83)

“A significant proportion of older adults are less likely to receive optimal cancer treatments. According to a prior study, only 30% of patients aged 80 years and older received chemotherapy, compared to 65% of those aged 18-59 years​.[4] However, recent studies suggest that older adults in good functional health can still derive meaningful survival benefits from active cancer treatment, particularly when treatment decisions are based on frailty rather than chronological age.[5] In fact, recent improvements in general health, driven by better nutrition, healthier lifestyles, [6] and expanded access to cancer screening, have contributed to earlier detection. This has led to a growing subset of older patients who are able to undergo and benefit from active treatment.[7,8]

  1. In summary, the manuscript presents scientifically interesting results. With the proposed improvements, it could significantly increase its overall quality. My recommendation is to proceed with a thorough revision as suggested, as the manuscript—if appropriately modified—could represent a meaningful contribution to the relevant scientific literature.

Response: We thank the reviewer for recognizing the manuscript’s potential and have conducted a comprehensive revision as suggested.

Reviewer 2 Report

Comments and Suggestions for Authors

Seung Tae Kim et al. reported an interesting cohort study in Korea. Cancer care is an important topic in the field of oncology, thus felling within the scope of Cancers. The manuscript was informative, and might add some value to its field. The reviewer would like to endorse a Minor Revision for this submission. Please refer to the detailed comments:

  1. The significance and meaning of the findings should be outlined at the end of Introduction.
  2. According to the Title, the patients aged ≥80 might be the focused group. However, according to the demographic data, only 13,111 out of 301,055 patients (4.36%) were aged ≥80. Were the findings robust? The reviewer supposed that the number of <80 and ≥80 age should be comparable.
  3. According to Table 1, over 24 types of cancers were diagnosed. However, in the later analysis, there were only 7 types of cancers studied. Please consider whether the other types of cancers should also be analyzed.
  4. Figure 1: The Figure Caption mentioned the timeframe as 2008~2021. But there were data on 2022 in the figure. Please double-check.
  5. The last paragraph of Discussion should be denoted as Conclusion.

Author Response

Response: We thank the reviewer for the thorough and insightful review. We have carefully addressed each of the specific comments and revised the manuscript to improve.

Specific comment

  1. The significance and meaning of the findings should be outlined at the end of Introduction.

Response: In response to the reviewer’s comment, we have revised the Introduction to clearly state the primary aim of the study, which was to descriptively evaluate trends in cancer diagnosis, treatment patterns, and survival outcomes among patients aged 80 years and older. In addition, we added a new concluding paragraph in the Introduction that outlines the expected contribution and significance of this study. We emphasized the relevance of our findings to geriatric oncology practice and the need for evidence-based policy development in aging societies. These revisions are reflected in the updated

Introduction (page 2, line 77-83).

“To address these gaps, we conducted a 15-year cohort study to analyze trends and survival outcomes by cancer type and stage in patients aged 80 years and older. This study aims to descriptively evaluate temporal trends in cancer diagnosis, treatment patterns, and survival outcomes in patients aged 80 years and older using a 15-year hospital-based cancer registry. These findings will contribute to a better understanding of how cancer care is changing in the population of very old adults, and could inform future guidelines and policies that support more individualized treatment decisions in rapidly aging populations over chronological age.”

  1. According to the Title, the patients aged 80 might be the focused group. However, according to the demographic data, only 13,111 out of 301,055 patients (4.36%) were aged 80. Were the findings robust? The reviewer supposed that the number of <80 and 80 age should be comparable.

Response: While patients aged 80 years and older represented 4.4% of the total cohort, this still corresponds to more than 13,000 individuals. This sample size provided sufficient statistical power to evaluate treatment trends and survival outcomes, including stratification by cancer type and stage. For example, assuming that 30% of patients received treatment and 70% did not, reflecting real-world proportions and assuming a 10% absolute difference in 5-year survival (e.g., 50% vs 40%), the minimum total sample size required to detect this difference with 80% power at a 5% significance level would be approximately 922 patients. Our cohort of 13,111 patients aged ≥80 years greatly exceeds this threshold, indicating that the findings are statistically robust. We have added this explanation to the Methods section.

Methods (page 3, lines 129-134).

“To evaluate whether the sample size of patients aged 80 years was sufficient for comparative survival analysis, we conducted a simplified power calculation. Assuming a 5-year survival rate of 50% in the treated group and 40% in the untreated group, with a treatment proportion of 30% and a significance level of 0.05, the minimum total sample size required to achieve 80% power was approximately 1000.

  1. According to Table 1, over 24 types of cancers were diagnosed. However, in the later analysis, there were only 7 types of cancers studied. Please consider whether the other types of cancers should also be analyzed.

Response: We thank the reviewer for this important observation. Although the overall sample size was sufficient for comparative survival analyses, we performed subgroup analyses only for cancer types with adequate case numbers in the ≥80 age group to ensure reliable and stable estimates. Cancer types with smaller sample sizes were excluded due to concerns about statistical instability. Therefore, we selected 7 major cancer types that were most prevalent in this population. We have now explicitly stated this rationale in the Methods section and acknowledged it as a limitation in the Discussion.

Discussion (page 12, line 283-287)

“Third, rare cancers were not analyzed separately due to small sample sizes. Nevertheless, compared to previous studies, our cohort included a substantially larger number of older patients with cancer, which enabled meaningful survival comparisons by treatment status even within this advanced age group.”

  1. Figure 1: The Figure Caption mentioned the timeframe as 2008~2021. But there were data on 2022 in the figure. Please double-check.

Response: We revised the typo in the Figure legends.

“Figure 1. Trends in the proportion of patients aged 80 year or more receiving treatment (2008-2022).”

  1. The last paragraph of Discussion should be denoted as Conclusion.

Response: In response, we have created a dedicated “Conclusions” section at the end of the Discussion.

Round 2

Reviewer 1 Report

Comments and Suggestions for Authors

The authors provided appropriate modifications to the manuscript